# All That Glitters Is Not Silver—A New Look at Microbiological and Medical Applications of Silver Nanoparticles

**DOI:** 10.3390/ijms22020854

**Published:** 2021-01-16

**Authors:** Paweł Kowalczyk, Mateusz Szymczak, Magdalena Maciejewska, Łukasz Laskowski, Magdalena Laskowska, Ryszard Ostaszewski, Grzegorz Skiba, Ida Franiak-Pietryga

**Affiliations:** 1Department of Animal Nutrition, The Kielanowski Institute of Animal Physiology and Nutrition, Polish Academy of Sciences, 05-110 Jabłonna, Poland; g.skiba@ifzz.pl; 2Department of Molecular Virology, Faculty of Biology, Institute of Microbiology, University of Warsaw, Miecznikowa 1, 02-096 Warsaw, Poland; mszymczak@biol.uw.edu.pl; 3Institute of Polymer and Dye Technology, Lodz University of Technology, Stefanowskiego 12/16, 90-924 Łódź, Poland; magdalena.maciejewska@p.lodz.pl; 4Institute of Nuclear Physics Polish Academy of Sciences, 31-342 Krakow, Poland; lukasz.laskowski@ifj.edu.pl (Ł.L.); magdalena.laskowska@ifj.edu.pl (M.L.); 5Institute of Organic Chemistry PAS, Kasprzaka 44/52, 01-224 Warsaw, Poland; ryszard.ostaszewski@icho.edu.pl; 6Moores Cancer Center, University of California San Diego, 3855 Health Sciences Dr., La Jolla, CA 92037, USA; 7Department of Clinical and Laboratory Genetics, Medical University of Lodz, 251 Pomorska Str., 92-213 Łódź, Poland

**Keywords:** silver, nanoparticles AgNPs, biofilm, antibiotics, eukaryotic and prokaryotic cells

## Abstract

Silver and its nanoparticles (AgNPs) have different faces, providing different applications. In recent years, the number of positive nanosilver applications has increased substantially. It has been proven that AgNPs inhibit the growth and survival of bacteria, including human and animal pathogens, as well as fungi, protozoa and arthropods. Silver nanoparticles are known from their antiviral and anti-cancer properties; however, they are also very popular in medical and pharmaceutical nanoengineering as carriers for precise delivery of therapeutic compounds, in the diagnostics of different diseases and in optics and chemistry, where they act as sensors, conductors and substrates for various syntheses. The activity of AgNPs has not been fully discovered; therefore, we need interdisciplinary research to fulfil this knowledge. New forms of products with silver will certainly find application in the future treatment of many complicated and difficult to treat diseases. There is still a lack of appropriate and precise legal condition regarding the circulation of nanomaterials and the rules governing their safety use. The relatively low toxicity, relative biocompatibility and selectivity of nanoparticle interaction combined with the unusual biological properties allow their use in animal production as well as in bioengineering and medicine. Despite a quite big knowledge on this topic, there is still a need to organize the data on AgNPs in relation to specific microorganisms such as bacteria, viruses or fungi. We decided to put this knowledge together and try to show positive and negative effects on prokaryotic and eukaryotic cells.

## 1. Introduction

Antibacterial properties of silver (Ag) have been known for millennia [1]. Already in ancient times, its features were used for storing food and water in silver vessels and healing wounds and ulcers with mixtures containing silver [2]. Silver has also been widely used as a universal therapeutic agent due to the wide spectrum of action against pathogens such as bacteria, viruses, fungi and protozoa. With the discovery of penicillin by Alexander Fleming in 1928 and the advent of antibiotics, the use of silver was limited due to the high costs associated with its production. Currently, the antibiotic therapy has become widespread. Excessive use of antibiotics, often inadequate to therapeutic needs in dentistry, run to the production of resistant bacterial strains like, e.g., *Bacillus amyloliquefaciens* [3,4]. It brings the need to look for alternative pharmacotherapy and methods to prevent infection. In the era of intensive development of science and modern nanotechnologies accessibility, the interest in silver’s role and properties in different organisms is back. 

### 1.1. The Characteristics of Silver Nanoparticles 

Nanoparticles (NPs) are characterized by dimensions ranging from 1 to 100 nm. Currently, two concepts of obtaining nanoparticles are distinguished. The first one called “bottom-up” is based on building them from scratch, atom by atom [5]. Depending on the desired properties of the final product, the substrates may be atoms, molecules and colloidal particles. 

There is also the reverse concept of producing nanoparticles, called “top-down”. It involves the fragmentation of the starting material, so that the desired product is in size not exceeding 100 nm.

Nanoparticles may have a diverse structure, be randomly arranged in space and can take on the crystalline internal structure [6]. They have been found in the form of monocrystals, or in a cluster of many crystals. There are also nanoparticles embedded in the structure of other metals [6]. Silver nanoparticles (AgNPs) have a diameter not exceeding 100 nm and consist of about 20–15,000 atoms [7,8]. Nanoparticles smaller than 10 nm penetrate cell membranes easier and faster [6,8]. The unique design of the Ag crystal unit cell (cube with 4 Ag atoms) allows oxygen adsorption, which can be released into the environment [6]. 

A characteristic feature of silver-containing nanoparticles in the analysis of their toxicity and biological and biomedical activity, their size, morphological structure, particle composition, shape, surface area distribution, solubility, aggregation on specific coatings, dissolution rate, reactivity of particles in solutions, effectiveness are taken into account but also release of specific ions, cellular uptake, biological distribution, penetration of biological barriers and the types of analyzed cells—Procaryota or Eucaryota [9,10,11]. To evaluate these parameters, very sensitive and specialized methods of their qualitative and quantitative analysis are used, which include analytical techniques such as visible ultraviolet spectroscopy (UV–VIS spectroscopy), X-ray diffraction (XRD), Fourier transform infrared spectroscopy (FTIR), X-ray photoelectron, spectroscopy (XPS), dynamic light scattering (DLS), scanning electron microscopy (SEM), transmission electron microscopy (TEM), atomic force microscopy (AFM) and nanoparticle flow cytometry using specific cell sorters [12,13]. 

### 1.2. Synthesis of Silver Nanoparticles

The synthesis of AgNPs can be carried out by six main methods: (1) chemical, (2) physical, (3) biological, (4) sonochemical, (5) physicochemical and (6) photochemical [14,15,16,17,18]. The morphology and stability of the nanoparticles can be controlled by the choice of synthesis physicochemical parameters, such as the concentration of silver salt and stabilizer or the molar ratio of the reducer to silver salt tested on pore structures of a virus [19,20,21,22,23,24]. Techniques of obtaining metal NPs are fast, inexpensive and environmentally friendly. It is well known that many microorganisms, such as bacteria, actinomycetes and fungi, as well as plants, are capable of synthesizing metal NPs [15]. Another way to obtain metal NPs is through peptides, which can be stabilizers of AgNPs [15,21]. Adhering to silver nanoclaves, peptides enhance the reduction of the silver ion and stimulate the growth of the crystal, reaching a size of 60–150 nm. Amino acids from peptides (free amino acids do not interact) reduce silver ions to silver crystal forms. Such properties include arginine, cysteine, lysine, methionine and tyrosine. The last one acts as a reducing agent in an alkaline environment [15,22]. The reduction of silver ion at high pH occurs due to the ionization of the phenolic group of tyrosine, which by reducing silver ions transforms into a semichinone structure. The tryptophan present in the protein is converted into a transient form which is an electron donor for the metal ion [15]. For the synthesis of AgNPs, coated with peptides, proteins containing disulphide bonds may be useful. The size of such NPs changes under the influence of environmental pH and aggregation process. This suggests that such a nanomaterial may be of interest for medical and biotechnological use. Greater understanding of the metal NPs mechanism formation makes this model more use to create a particle of the plant virus for gold NPs synthesis. Tyrosine residues on the viral capsid reduced AuCl_4_, forming gold NPs, deposited on the surface of the virus [16,17,18,19,20,21,22,23,24].

### 1.3. Physico-Chemical Properties of Silver Nanoparticles

The unique physico-chemical properties of AgNPs as shape, size and change in melting point with respect to metallic silverare associated with a large ratio of an active surface to a volume [18,19,20,21]. Based on this relation, it is possible to achieve high antimicrobial effects by using very low silver concentrations (ppm). Silver NPs, applied as antibacterial, anti-fungal and anti-viral compounds, have various shapes like spheres, tubes and plates [11,12,14,16,22]. It is speculated that a triangle or plates-forms have the strongest activity due to a larger active surface [22]. To reduce materials to the nano scale, a complete change in the energy of the system is needed. This is related to the disturbance in that material’s thermodynamic properties. Silver nanoparticles with a diameter of 2.4 nm melt at an approximate temperature 360 °C, while unground silver without additives melts at 961.9 °C, however, boils at 2210 °C [17,18,19]. In addition, its physical properties such as thermal, electrical and magnetic conductivity, also vary with the size of the silver nanoparticles. Silver alloy is easier to melt than pure metal ingots because the impurities lower the melting point [17,18,19]. The optical, adsorptive and reactive properties, which are closely related to the increased antimicrobial activity of AgNPs, also change with the particles size [14,16]. In this discussion we mainly focus on biological properties of AgNPs [14,17].

## 2. Nanomaterials as New Potential Antibiotics

Lately, the growing interest of pharmaceutical companies in the search for designing and marketing of new antibiotics has resulted in the intensive research and development of new treatments [22,23,24,25,26]. Potential possibilities of replacing typical antibiotics are based on the use of antibacterial peptides, probiotics, virulence inhibitors of pathogenic strains and antibodies. Moreover, compounds of vegetable origin, bacteriophages and their lytic enzymes or the latest products of nanotechnology, biogenic nanosilversilver, or gold, copper and platinum NPs have been used against selected microorganisms like multidrug-resistant bacteria (MDRB) [26]. New promising achievements in nanotechnology allow combating microbes. There are several technologies to synthesize NPs and nanocomposites (e.g., peptidomimetics) with strictly defined physicochemical properties and significant biological potential [22,23,24]. It allows the nanodrug directly access the specific organ, which reduces side effects and facilitates its delivery inside the cells. Among practical applications, the most popular are water nanocolloids, i.e., particles dispersed in water. This is an unstable system, due to increasing concentration, drastic changes in pH and temperature occur [25,26,27].

In order to obtain a stable colloid, compounds preventing aggregation and sedimentation are used, such as citric acid, tannic acid, chitosan, methyl cellulose, ethylene glycol or polyvinylpyrrolidone (PVP) [27]. As with other nanomaterials (NM), the features that determine the physical properties of AgNPs are the ultra-small dimensions that affect the ultra-large surface in relation to the mass in which a significant proportion of atoms are in direct contact with the environment and are ready to enter with it in reactions with cell free or cell-associated HIV-1 [28]. Through various synthesis methods, it was possible to create nano forms with spherical, longitudinal or triangular shapes, which also effectively impairs antibacterial activity. It turned out that silver nanoparticles in the shape of trimmed triangular flakes have the strongest bactericidal and antifungal potential in dental implants [29,30,31].

### 2.1. The Antimicrobial Properties of Silver and Silver Nanoparticles

The antibacterial activity of silver and its compounds has been known for a long time. Already in the 19th century, silver salts and its colloids were commonly used in the treatment and prevention of infections caused by microorganisms, also in sepsis, acute epididymitis, mig-donors and in infants’ conjunctivitis. After many years of oblivion, silver returns to its glory. In the last decade, it has been seen a silver large coming back in the form of nanoparticles, mainly used as carriers for drugs and other active substances in cosmetology and cancer therapies. Ionic silver was replaced by nanoparticle form due to its susceptibility to complexing and precipitation. It is also suggested that AgNPs cause fewer side effects, and their safety and antibacterial potential can be further enhanced by various modifications, such as an addition of stabilizers [30,31,32,33]. It has also been shown that AgNPs have an inhibitory effect on various microorganisms (including *Escherichia coli*, *Salmonella enteritidis*, *Listeria monocytogenes*, *Pseudomonas aureginosa*, *Staphylococcus aureus*, *Candida albicans*), which was determined in vitro at around 10–15 ppm [9,30,31,32,33]. The same AgNPs enriching drinking water for quail and poultry did not inhibit the number of most gastrointestinal bacteria; however, an increase in the number of lactic acid bacteria was observed [4,24,26,32]. This mechanism of biointeraction between nanoparticles and bacteria was visualized in an optical and confocal microscope [32]. Observations carried out for several years on industrial poultry farms, where AgNPs were added to drinking water at the level of 1 ppm, showed significantly lower incidence of chickens and greater weight gain [34,35]. Although it occurs in small amounts, it determines many processes in the human body causes immunity impairment [36,37,38]. Protracted industrial exposure to silver can lead to arteriosclerosis, kidney and liver damage [36,37,38]. The American Society of Governmental Industrial Hygienists made a regulation of a safety daily exposure for humans and for metal-silver the safety dose is 0.1 mg/m^3^, whereas for the colloidal form is 0.01 mg/m^3^. WHO has determined that the acceptable concentration of silver in drinking water should not exceed 0.1 mg/L, while in food it cannot be higher than 0.05 mg/kg [10,36,37].

Silver NPs seem to act nonspecifically, eliminating microorganisms from the environment, but on the other hand, some bacteria (*Pseudomonas species*, *Klebsiella pneumoniae*), fungi (*Verticullium*, *Fusarium oxysporum*, *Aspergillus fumigatus*) and non-pathogenic fungi (e.g., *Trichoderma asperellum*) are able to synthesize NPs from silver salts present in the surrounding environment [38,39,40,41,42]. It is believed that the biosynthesis ability of NPs from dissociated silver salts protects these microorganisms against a more toxic ionic NPs form. Recently, the adaptation process of *Bacillus subtilis* to intracellular oxidative stress caused by long-term exposure to AgNPs has been found [4,39]. Metallic silver on the nano scale inhibits the growth and development of both Gram-negative bacteria (*Escherichia*, *Pseudomonas*, *Salmonella* and *Vibrio*) and Gram-positive bacteria (*Bacillus*, *Clostridium*, *Enterococcus*, *Listeria*, *Staphylococcus* and *Streptococcus*), including strains resistant to antibiotics (e.g., *Acinetobacter*, *Staphylococcus aureus* and *Enterococcus faecium*). Nanoparticles (average size 22 nm), combined with some antibiotics, may increase their efficacy [3,4,5,11]. The bactericidal effect of silver is related to the mechanism of thiol groups interaction with the bacterial cell wall, increasing membrane permeability, causing ionic disorders and destructive impact on DNA interaction. Under the influence of silver, metabolic products are accumulated inside the cell and the possibility of protein synthesis is suspended [4,5,11]. The multidirectional interaction of NPs with a bacterial cell leads to its death. The mechanisms of induction of cell death by AgNPs are observed in *E. coli* bacteria, especially in R1-R4 rough strains having equal LPS length, reduced cellular components such as sugars and proteins [43,44]. Moreover, AgNPs, like peptidomimetics, are capable of destroying the structures of bacterial membranes by increasing its permeability and creating many micropores, penetrating inside the bacterial cell, inducing oxidative stress [45,46]. During this process, the DNA structure is destroyed by oxidation and alkylation of bases, leading to the formation of such modifications as 8-oxoguanina, 7,8-dihydro-8-oxoguanine (8-oxoguanine), 8-oxoadenine, unsubstituted and substituted imidazole ring-opened purines introduced into DNA by hydroxyl radicals (e.g., FapyG, FapyA), as well as by chemical carcinogens, including anticancer drugs (e.g., Fapy-7MeG, Fapy-7EtG, Fapy-7aminoethylG, aflatoxin B1-fapy-guanine, 5-hydroxy-cytosine and 5- hydroxy-uracil), fapy-adenine (FapyG) and fapy-guanine (FapyA) [45,46]. In the analyzed *E. coli* strains, it was also observed that smaller AgNPs penetrating into the cell membrane having a larger surface and shape showed more effective antibacterial activity than larger AgNPs [45,46,47,48,49,50]. So far, bacteria are unable to develop adequate defense mechanisms against AgNPs. The effective action of silver, gold, copper and platinium against such pathogens as *S. aureus*, *E. coli*, *P. aeurginosa*, *A. baumani*, *E. feacalis*, *S. epidermidis*, *C. albicans*, *Streptococcus mutans*, *Salmonella* and *Campylobacter* strains and peptidomimetics are documented [47,48,49,50,51,52,53,54]. 

Silver nanoparticles, coating materials for medical applications, inhibit the secretion of bacteria by the polysaccharide extracellular matrix, i.e., biofilm, which significantly improves the activity of the immune system and antibacterial agents of endo- and exogenous origin. Silver for a long time has been showing antibacterial activity and this effect is observed even at low concentrations [47,48,49,50,51,52,53,54]. Nanda et al. [54] reported that the growth of *E. coli*, *S. aureus* (MRSA), *Providencia*, *Serratia* and *P. aeruginosa* is inhibited by the presence of ~1 g/mL silver ions and *Corynebacterium pseudotuberculosis* infection in small ruminants [54]. Bacterial resistance to silver can be associated with the accumulation of nanoparticles in the intercellular space, e.g., in bacteria of the genus *Morganella* [55]. Such bacteria can reduce metal due to increased resistance to toxicity. This feature may be useful in bioremediation [56]. 

In vitro study confirmed the antibacterial properties of silver nanoparticles in animals in relation to pathogens of poultry gastrointestinal tract, including *E. coli*, *Salmonella enteritidis* [36].

The mechanism of antibacterial action of AgNPs has not been fully explained; however, scientists agree on the hypothesis that AgNPs are able to anchor and penetrate bacterial membranes due to interaction with surface proteins, ultimately leading to structural changes in the protein-lipid layer, disruption of transport and cell death. It has been found also that the proposed mechanism of killing microorganisms is associated with the formation of free radicals that trigger the cascade reaction of lipid peroxidation and membrane destabilization [56,57]. At the same time, it has been shown a greater tolerance to AgNPs by Gram-positive bacteria, whose cell wall consists of many layers of peptidoglycan-murein, providing a strong negative charge. Probably this physicochemical property of the bacterial wall inhibits silver contact with the cellular cytoplasmatic membrane. The ions of this element released on the surface of NPs are also responsible for the antimicrobial effects of metallic silver, according to the principle: the smaller the nanoparticles, the more important the silver ionic form is in inhibiting bacterial growth [50,57]. The metal cations present in the cytoplasm are not effectively removed from the cell as a result of damage caused by NPs. Silver in ionic form reacts with thiol groups of peptides and enzymatic proteins as well as phosphorus-containing bases, disrupting cell metabolism and also causing oxidative stress [30,58].

Further damage can be caused by the interaction of AgNPs with DNA and inhibition of cell division and DNA replication. In addition, it was hypothesized that inhibition of bacterial growth could take place in the pathway of interaction of AgNPs with phosphotyrosine that is part of bacterial peptides responsible for signal transduction in the cell. AgNPs are capable of modulating the molecular pathways of *Morganella* cells, as demonstrated in studies [55,56,57,58,59,60]. Numerous studies indicate the anti-inflammatory properties of AgNPs, associated with downregulation of TNF, IL-12, IL-1, NF-kB and induction of apoptosis [59,61]. In addition, a silver modulating role in activity of cytokines involved in wound and periodontal pathogens healing process was found [51,62,63]. 

### 2.2. Bacteria as a Target for Different Forms of Silver

Bacteria have been causing many diseases for thousands of years. At the beginning people were unable to deal with them until the invention of antibiotics. The first was penicillin, which was able to inhibit the division of bacterial cells by blocking synthesis of muramic acid. Unfortunately, not all bacteria are sensitive to penicillin and other antibiotics [57,64,65]. The efficacy of various currently used antibiotics such as amoxicillin, erythromycin, clindamycin and vancomycin methicillin against various pathogenic bacteria such as *S. aureus* (MRSA), *S. epidermidis*, *E. coli*, *Streptococcus pyogenes*, *Salmonella typhi* and *Klebsiella pneumoniae* were grown in the presence of AgNPs [30,54]. Chitosan nanoparticle combined with silver plays a similar role as the described antibiotics, because its synthesis into the Ag polymer promotes the formation of small AgNP which can be added to selective media with a pH range of 6.3–6.5 in the form of silver hydrogel nanocomposites [66,67,68,69].

It is known that metals are toxic to most microorganisms, but silver, for example, does not exert any influence on some bacteria. However, silver nitrate in higher concentrations was used long time ago as a wound disinfectant and inflammation agent. It has been used in many forms so far, among others to disinfect the eyes of newborns, mainly to eliminate pathogenic *Pseudomonas* and *Staphylococcus* strains [38]. Another important compound used due to its valuable antimicrobial properties is silver sulphadiazine and complexes of this element with compounds from the group of sulfonamides. Formerly it was used to disinfect postoperative wounds or ulcers. They are more effective than silver nitrates, thanks to the slower release of silver ions into the reaction environment, which prevents the cations from sticking together and the formation of ineffective forms of this element. Currently, there are many limits on the production, use and consumption of AgNPs, due to the lack of sufficient evidence and analyses regarding the risks associated with them [70]. 

### 2.3. Production of Silver Nanoparticles by Bacteria 

The laboratory production of AgNPs by bacteria is of interest due to the uncomplicated procedure, the ability to modify the process and enhance the effect using genetic engineering. The first report on the biosynthesis of AgNPs concerned the microorganisms isolated from cases of mastitis in cattle [47]. The NPs produced by this bacterium formed single crystals of a specific shape, that were accurately characterized using a transmission electron microscope (TEM), X-ray dispersion energy (EDX) and electron spectroscopy diffraction (EDS) [12,13,14]. The silver-containing crystals were surrounded by the organic substance of the bacteria; they reached a size of up to 200 nm and had a triangular or hexagonal shape. Synthesis of AgNPs were observed by bacteria of the genus *Morganella* sp. This process was initiated after 1 h and increased up to 18 h when 90% of NPs with a spherical shape were produced [53]. The extracellular synthesis of AgNPs (5–50 nm) in the presence of silver ions was also observed in *Bacillus subtilis* cultures [4,39]. The NPs were stable, they did not aggregate, and stability was provided by protein coating. The liquid after culturing these bacteria showed high activity of nitrate reductase. This enzyme, along with electron transporters (NADH) and other proteins, may be responsible for the reduction of silver ions and subsequent formation of nanoparticles in most bacteria. Nitrate reductase induced by nitrate ions reduces silver ions to metallic silver [29,71]. The nanocrystalline complex of chlorhexidine Ag (III) CHX silver and its ligand (CHX) and silver sulfadiazine show similar antibacterial activity to the tested Gram-positive or -negative strains [50,57,72]. Likewise, the AgNP complex in combination with bioactive guava leaf components (Gr-Ag-NP) showed significantly higher antimicrobial activity and stability against *E. coli* compared to chemically synthesized AgNP. The reason for this higher activity may be due to the adsorption of biomolecules on the Gr-Ag-NP surface. Moreover, AgNPs synthesized by *Cryphonectria* sp. showed antibacterial activity against various human pathogenic bacteria, including *S. aureus*, *Klebsiella pneumoniae*, *E. coli*, *Salmonella typhi* and *Candida albicans* [9,30,73,74,75,76,77,78,79].

### 2.4. Mechanisms of Silver Particles Transport to Bacterial Cells

Bacteria have several mechanisms that allow them to absorb silver particles into the cells. There are two main ways they produce metal ions. The first one is based on the concentration gradient production on both sides of the cytoplasmic membrane [35]. It is characterized by a lack of specificity and runs at a very high speed, causing oxidative stress that induces bacterial DNA damage (Figure 1).

The second transport type of metal ions is much slower, specific in relation to metals and requires energy input in the form of adenosine triphosphate (ATP) (Figure 1) [35]. However, there are other mechanisms for the collection of metal ions by microbial cells. In Gram-positive bacteria, the glutamic acid groups and the teichoic phosphodiester groups show very strong silver ion binding. In contrast, Gram-negative cells bind metallic silver ions on the outer peptidoglycan protein layers [50]. Fungus cells are also not completely resistant to silver [40]. Nanosilver destroys the wall and cell membrane in the same way as in bacteria, inhibiting cellular respiration and deactivating enzymes [50]. It has been proved that bacteria exposed to the direct action of AgNPs are destroyed respectively in 99.99% and Eukaryotic cells of fungi in 100% [40]. Nicotinamide adenine dinucleotide (NADH)-dependent nitrate reductase and quinone transports participate in the synthesis of fungal AgNPs. Nanoparticles operating on bacteria are stabilized by proteins [39,40]. *Trichoderma asperellum* cellular extract analysis, after removal of the produced AgNPs, showed the increase of II amine bonds, O-H and S-H bonds as well as C=O carboxylic and carboxylic bonds indicating an increase in protein concentration in the nanoparticles after their formation (Figure 1) [39,40,74,75,76,77].

### 2.5. Nanoparticles Inhibit the Growth of Bacterial Biofilms

Biofilms are common in various environments, both natural and created by man. They arise on almost all humid surfaces and inhabit almost every living organism. Cells included in the biofilm differ physiologically and morphologically from the free-living cells of the same species. The complex process of biofilm formation consists of several characteristic stages, such as adhesion, formation, maturation and its decay [77,78,79,80,81]. Biofilms are settled populations of one or more species of microorganisms that are attached to a biotic or abiotic solid surface or are formed at the interface [77,78,79,80,81]. Bacteria living in biofilms can easily spread in the body, leading to the chronic and difficult to cure infections. This is facilitated by the slow growth of cells and their ability to avoid host’s immune system attack [77,78,79,80,81]. Cells in biofilms are surrounded by extracellular polymeric substance, mainly composed of polysaccharides but also from proteins and nucleic acids, called a biofilm matrix. Many of bacterial species coexist in the microbial communities, e.g., *P. aeruginosa* and *S. aureus* [31,38]. The disturbance of their functioning by silver nanoparticles leads to changes in the virulence of these bacteria. Over 65% of all nosocomial infections come from bacterial biofilms, mainly formed by *E. faecalis*, *S. aureus*, *S. epidermidis*, *E. coli*, *K. pneumoniae* and *P. aeruginosa* [31,38,77,78,79,80,81]. The ability to create biofilms is of direct importance in the pathogenesis of cystic fibrosis, gingivitis and middle ear otitis [31,38,77,78,79,80,81]. 

It is well known that cells in biofilms are characterized by increased resistance to therapies compared to slow-moving cells, who tolerate the presence of up to 1000-fold higher doses of toxic compounds, e.g., antibiotics [26,57,64,65]. The observed antibiotic resistance of bacteria in biofilms is the effect of many mechanisms. The presence of a matrix, which is a barrier limiting the penetration of the antibiotic into the biofilm structure should be mentioned as first and it applies specially to mature, extensive biofilms [31,38]. Extending the penetration time of antibiotics allows the bacteria to activate other protective mechanisms, such as production of enzymes, e.g., β-lactamases catalysing the hydrolysis of β-lactam antibiotics [26,57,64,65]. Secondly, the existence of a significant fraction of persistent cells, whose features are discussed above, can cause the resistance. Finally, the effective functioning of multidrug transporters, MDR pumps, present in pathogenic bacteria *P. aeruginosa* and *S. aureus*, are of big significance [26,57,64,65]. The resistance of biophilic pathogenic bacteria means that effective fight against them is one of the greatest challenges that modern medicine has to meet. Microscopic observations have shown that nanoparticles, when attacking cell membranes, are able to change cell morphology [56].

Research on the potential of AgNPs in combating biofilm, perfectly meets the expectations of medicine [7,66,80]. It is known that nanosilver efficiently prevents the formation and further stages of biofilm development; however, the effect depends on the species and even the bacterial strain. This effect is generally stronger against biofilm formed by Gram-negative pathogenic bacteria [50,57]. There are also reports regarding AgNPs inhibition of biofilm formation on medical devices. Urological catheters with nanosilver surface were shown to be resistant to colonization by *E. coli*, *S. aureus* and *C. albicans*, even under conditions of continuous fluid flow [31]. To achieve this goal, various stabilizers, such as starch, citrate, amino-silica and ZnO particles in a composite resin, are tested [82] (Figure 2). 

Biofilms not only lead to antimicrobial resistance but are involved in their development eye-related infectious diseases such as bacterial keratitis [38]. Interestingly, Khurana et al. (2014) [83] investigated the antimicrobial properties of silver nanoparticles based on its physical and surface properties against *S. aureus*, *B. megaterium*, *P. vulgaris* and *S. sonnei*. The enhancement of the antimicrobial activity was observed for particles with a hydrodynamic size of 59 nm compared to 83 nm. Nanocomposites containing graphene oxide (GO)-Ag, graphene alone and AgNPs showed increased antimicrobial activity against *E. coli*, *S. aureus*, *P. aeruginosa*, *A. baumannii*, *E. faecalis* and hospital bacteria of the MRSA type, using the conventional disc diffusion method [54,84,85,86].

## 3. The Effect of Silver on More Developed Organisms

The presence of nanotechnology products, both in the industry and in everyday applications, increases the number of studies on the effect of AgNPs on cells of more developed organisms. Potential toxicity of nanoparticles and their effect on growth, development and health were investigated in studies conducted on leukemic cell lines [87], the chicken embryo model and broiler chickens [34,35]. In vitro studies showed a toxic effect of AgNPs on keratinocyte cells, triggering apoptosis, while a higher percentage of necrotic monocytes was observed [88]. The use of silver, gold, copper and palladium nanoparticles at a level of <100 ppm, administered to the albumin or egg air chamber, did not adversely affect the growth, development and health of birds. AgNPs exerted no effect on oxidative DNA degradation in the muscles of hen embryos treated in ovo with nanoparticles. Moreover, AgNPs were found to stimulate the immune response by increasing the level of IgG in chicken embryos [20,21,47,66,87,89,90,91]. Administration of up to 25 ppm of AgNPs in drinking water did not negatively affect the state of quail health, as well as the morphology of bird enterocytes [89]. Usually, the first step of application testing is completed on the cell lines before moving to animal testing. The cytotoxic effect of AgNPs on rat and mouse cells, including adrenal tumor cells A549, PC-12 and hacat cells, has been proved [92,93,94]. Relatively, many studies concerned the impact of AgNPs on human cells, including macrophages, human fibrosarcoma cells (HT-1080) and epithelial cells (A431) [29,93]. It was proved that AgNPs induced secretion of proinflammatory cytokines, such as interleukins Il-6, Il-10 and TNF-α in macrophages; however, in A431 cells after the exposure to AgNPs, oxidative stress, elevated lipid peroxidation and DNA fragmentation were observed, suggesting an induction of apoptosis. The effect of nanoparticles on the cell lines A549 and SGC-7901 (gastric cancer), HepG2 (liver cancer) and MCF-7 (breast cancer) was also tested [87,92,93,94]. Higher organisms are exposed to nanoparticles mainly through their respiration and contact with the skin. The human body is additionally endangered by application to wound surfaces or biomaterial implantation. It was proved that AgNPs delivered to the body through the respiratory system, digestive system or injection were subsequently identified in blood of animals, and after some time, they exerted a toxic effect on several organs, including the brain. Reductions in lung performance and increased inflammation, following by small changes in liver, have been observed. On the other hand, AgNPs applied on the surface of the guinea pig skin caused the local inflammatory reaction and penetrated deeper parts of the body, causing slight damage to the liver and spleen. In *Danio rerio*, a delay in egg hatching, body axis disorder, dorsal strand and cardiac arrhythmias were noted, while in mice, a fibrosarcoma was observed [95]. Inhalation of AgNPs is the most dangerous threat to human health. Metallic silver particles, from the bronchi, penetrate into the bloodstream and then directly to the internal organs [96]. A large amount of AgNPs causes damage to the mucous membrane of the upper respiratory tract, and in extreme cases, even fatal pneumonia. Silver nanoparticles from the circulatory system get into the liver, where they react with metallothionine [97]. Impaired enzyme function may result in an increase in hepatic transaminase levels, decreased number in erythrocytes and human skin fibroblasts [98] and enlargement of the liver and spleen. It is also believed that silver may have an allergenic effect. Excess of metallic silver particles are excreted via kidneys, digestive system and bile. It is noted that clinical symptoms, such as wound exudate, lower the cytotoxicity of AgNPs, and in this case, the high concentration of proteins in the exudate may neutralize the toxicity of nanosilver to neighbouring tissues (Figure 3) [1,14,56,94,99,100,101,102,103].

### 3.1. AgNPs May Induce Cellular Stress in Tissues of Developing Chicken Embryos

The kinetics of the reaction of metallic silver with the changing environment of the bacterial biofilm of gastrointestinal tract is still uncharted waters. It is not excluded that a wide range of pH (2–8), a mixture of many chemical compounds, microflora and immune cells effectively suppress the desired properties of NPs. Therefore, stable polysaccharide matrices have been used, to make AgNPs more effective, e.g., in contact with epithelial cells or with microflora along the entire length of the gastrointestinal tract. No AgNPs influence on morphological and biochemical indicators of peripheral blood of 18-day-old chicken embryos was found [34,35], and it was proved that Ag particles deposited in bones did not affect their physical properties, including tensile strength and hardness. Hedayatia et al. (2019) showed that AgNPs administered by injection into the “egg” increase the uptake of zebrafish embryos in early embryogenesis, which may indicate more efficient use of fat during this period and saving carbohydrate reserves [104]. There is no evidence for negative production effect of NPs delivered by injection in ovo, based on the assessment of body weight and muscle mass of zebrafish, immediately after hatching and after 28 days of rearing [104]; however, an improvement in the morphological structure of the pectoral muscle was observed on the day 20 of embryo incubation. Probably, the lack of a production effect could have been caused by post-translational modifications of vascular endothelial growth factor (VEGF) as FGF, which under the influence of AgNPs has been expressed only at the mRNA level [92]. AgNPs may induce cellular stress in tissues of developing chicken embryos. The mechanism of action depended largely on the type of cells that can activate both apoptosis and necrosis in vitro [34,35]. The data revealed also the anti-inflammatory potential of silver nanocolloid [27]. In case of embryos inoculated against Gumboro disease, nanoparticles increased the expression of HSP70 heat shock proteins and increased the concentration of IgG antibodies in serum, which is a good predictor in the use of silver as an adjuvant component of poultry virus vaccines [61,101]. Increased concentration of AgNPs with a fixed concentration and diameter may cause neurotoxic effect in vitro and in vivo. The data show that the basis of the AgNP threat to animal and human tissues and cells are disorders of the redox system, reduced concentration of glutathione, degradation of thiol groups in cytosolic proteins and cell membranes, disorders of mitochondrial activity and destabilization of cell membranes and astrocyte network [61,101].

### 3.2. Antiviral Effect of Silver Nanoparticles 

Strong inhibitory properties of AgNPs toward herpes simplex virus (HSV) have been reported [88,103]. That report opened the door to the use of AgNPs in the treatment of another viral infections. Experimental work conducted on HSV by two teams of scientists from Poland at the Warsaw University of Life Sciences (SGGW) and the Faculty of Chemistry of the University of Lodz, concerns the antiviral effect of AgNP of 13, 33 and 46 nm size, stabilized with tannic acid that is known from anti-inflammatory properties. Tannic acid, as one of the components in the synthesis of various AgNP nanoparticles of various shapes and sizes, was able to reduce HSV-2 infections both in vitro and in vivo by directly blocking virus attachment, penetration and further spread [105,106,107,108,109,110]. This construct effectively inhibits the penetration of HSV into target cells (unpublished data). The aim of the study was to determine their effect on the dangerous genital herpes virus—HSV-2. Tests carried out on keratinocyte cell lines revealed that nanoparticles of 13 and 33 nm completely inhibited that viral infection. The virus, passing through mucous membranes and inhabits a nerve ganglia, causes relapses of genital HSV. It happens in 10–60% of population, in developing countries even among 80% of HSV-2 infection elevates the risk of another infection leading to sexually transmitted diseases, human immunodeficiency virus (HIV) in particular [106,107,108,109,110]. Newborn babies infected with HSV-2 during a childbirth can develop meningitis and brain malformations. The HSV mouse model research reflects the course of diseases in humans. Tissue sections showed a significant reduction in the number of HSV-2 foci and their size, which confirmed their antiviral activity [104,105,106,107,108]. Biologically synthesized AgNP inhibited the survival of the herpes simplex virus (HSV) types 1 and 2 and the human parainfluenza virus type 3 based on the measurement of the size and zeta potential of the analysed AgNPs [106,107,108,109,110,111,112]. 

Similar studies are currently underway against SARS-CoV-2 evoking COVID-19. This coronavirus causing a life-threatening pneumonia exploded in December 2020 in the city of Wuhan in central China and spread to all countries of the world. The number of infected people and fatalities is counted now in the millions worldwide. In Europe, at the beginning of pandemic, the fastest growth of contracted coronavirus was recorded in Italy, where the number of deaths surpassed quickly those in China. In the vast majority of infected, the course of disease is quite mild, sometimes it may not even give symptoms. COVID-19 poses the greatest threat to the elderly and those with preexisting conditions. The high level of viral infection is associated with the remarkable ability of viruses to mutagenesis. It allows them to regain advantage over the host by creating new viral strains with acquired resistance to most of the available antiviral medicines. AgNPs of a size close to the size of the virus (20 nm) have an antiviral effect, by penetrating its nucleic acid and destroying or modifying its structure, which prevents a further self-replication [97,111,112,113,114,115,116,117,118,119,120,121,122,123,124,125,126,127,128,129,130,131,132,133,134,135,136,137,138,139,140,141,142,143,144,145,146,147,148,149,150,151,152]. The course of many viral infections is regulated by very complex interactions between the virus particles and a specific protein synthesis system as a defense of host cells. All viruses replicate through a similar sequence of events. AgNPs-based viral vaccines can be very effective against HIV-1, malaria, hepatitis B syncytial respiratory virus, HSV-1, monkey pox virus, influenza virus and Tacaribe virus [28,105,113,114,115,116] (Figure 4).

The antiviral properties of a combination of 50 ppb Ag with 20 ppm CO_3_ incorporated into ultrafiltration polysulfone membranes (nAg-PSf) against MS2 bacteriophage over a period of 15 min showed that such a complex has strong antiviral properties through increased membrane hydrophilicity [113]. AgNPs also demonstrated inhibitory activity against HIV and hepatitis B virus (HBV) [111,112]. 

### 3.3. Toxicity of Silver Nanoparticles in Tissues

Most of the toxicity studies of nanoparticles have assessed the survival of cells exposed to these nanoparticles. DNA toxicity has been the subject of a few of them, a phenomenon that does not necessarily lead to cell death but can lead to cancer mutations that, if not repaired by the cell itself, can lead to cancer. These nanoparticles at a concentration of 10 µg/mL—interestingly, the insufficient dose to kill all cells—caused the biggest number of damaged single-strand DNA. One of the concerns is an exposure to nanoparticles in the workplace. Babies and fetuses are also potentially at higher risk as their cells divide more often, making their DNA more susceptible to damage. Most often, nanoparticles enter the human body through the skin, lungs and stomach. Therefore, scientists are now focusing on studying the effects of nanoparticles on the DNA of these cells. They also investigate the effects of other than aforementioned nanoparticles, including metal oxides used in printer and photocopier toners, which can enter the air and that way make the lungs exposure. Model studies on Gram-positive and Gram-negative bacterial cells (Figure 1) seem to prove this statement [50,57].

A recent study at the Massachusetts Institute of Technology (MIT) and Harvard School of Public Health (HSPH) indicated that some nanoparticles can damage DNA [117]. Scientists have found that AgNPs added to toys, toothpaste, clothes and other products, due to their antibacterial effect, also severely damage nucleic acid molecules. They found that zinc oxide, i.e., nanoparticles very often used in sunscreen creams to stop ultraviolet rays, significantly destroys DNA. The results are based on the high-speed screening technology by which the degree of DNA damage was examined. That method allowed researchers to investigate the potential impact of nanoparticles at a much faster pace and on a larger scale than before. The US Food and Drug Administration (FDA) and European Commission–Joint Research Centre does not require manufacturers to test added nanoparticles if the substance has been shown to be harmless (e.g., it does not require AgNPs if silver itself is considered to be harmless) [36,118,119,120]. However, there are indications that some substances used in the form of nanoparticles, including those with the addition of silver, may be dangerous due to the fact that they are extremely small, have different physical, chemical and biological properties and the fact that they get inside human cells more easily [121]. In addition to silver, scientists analyzed four types of industrial-scale nanoparticles: zinc oxide, iron oxide, cerium oxide and an allotropic form of silicon dioxide (known as amorphous silicon). Some of these nanomaterials produce free radicals (reactive oxygen species, ROS) which are capable of altering the structure of DNA. As soon as these particles enter the body, they can accumulate in the tissues, causing even more damage. To be able to estimate a risk on humans exposed to metal oxide nanoparticles, more research needs to be determined checking the dose and the time of exposure initiating DNA damage [65,121,122,123,124,125].

Documented knowledge about the toxicity of AgNPs and their ability to accumulate in tissues (depending on the experimental model) strongly limits the application option, in which long-term contact of animals with this element in nano form is required. The lack of standardization criteria in the evaluation of nanopreparations used in experiments, might be also a problem. Hyung-Geun Park et al. (2017) proved the toxic effect of commercial silver nanocolloid (Nanocid) at the concentration range of 5 to 100 mg/L in hydra cells. The hydra cells have a strong regenerative capacity about toxic chemicals, excellent adhesion, aggregation and self-organization stages. The hydra regeneration test examined two types of silver nanocolloids (Ag NCs) and silver nanotubes (Ag NTs) at concentrations of 5, 10, 50, 100 mg/L, respectively [27]. The presented results confirmed a dose-dependent toxic effect of nanoparticles (hypoxia, lethargy, swimming disorders, acceleration of gill ventilation); however, the authors found that colloid is more economical and safer than commonly used high-dose disinfectants, e.g., malachite green, formalin, copper sulfate, chloramine. It is difficult to assess the scale of the risk associated with the release of nanoparticles, that can be used in aquaculture, into the aquatic environment. Food and feed safety are an important element of the internal market. The conditions for monitoring the quality of the product range placed on the market must include a range of measures to guarantee the health and general well-being of consumers. For that reason, feed additives and food in general, before being placed on the market, undergo the European Union (EU) safety assessment, based on a registration decision [120].

The European Food Safety Authority (EFSA), supported by the EU Reference Laboratory for Feed Additives, is responsible for assessing the registration application. In 2009, EFSA issued an opinion on the impact of nanomaterials (NM) on food safety, feed and related risks to humans and animals. The final provisions of the document draw attention to the still imperfect parameterization system (characteristics, qualitative and quantitative assessment) of NPs in food, feed and biological matrices. EFSA was concerned about insufficient information on the toxicokinetic and toxicology of nanomaterials (especially in oral contact) and their effect on the environment (Figure 5) [118,119,120]. 

## 4. Conventional Silver Preparations Used in Medicine

The most popular forms of silver such as silver nitrate, silver salt of sulfadiazine and other combinations of sulfonamides with silver have found applications in such fields of medicine as dermatology, urology, dentistry, ophthalmology and gynecology [80,108]. Nanosilver-containing burn wound dressings (Acticoat, Actisorb Plus) are characterized by longer antibacterial activity and better penetration deep into necrotic tissues compared to traditional silver compounds. In a study carried out at the Burn Treatment Center in Siemianowice Śląskie (Poland) in the years 2005–2006 on a group of 45 burned patients, more than 50% reduction in bacterial flora isolated from the wound have been obtained, regardless of its healing stage. Among the eliminated strains, bacteria responsible for the development of sepsis (*P. aeruginosa* and *A. baumani)* have been found [51,62,63]. Silver has antimicrobial and antifungal properties only to the ionic Ag+ form, which is unstable and can transform into an AgO form devoid of medicinal properties. Inactive silver (AgO form), which can accumulate in the tissue during long-term treatment, can cause skin and mucous membrane damage. It can also delay wound healing process due to tissue granulation. This is probably the reason for the different approaches to assess the clinical efficacy of many medical devices and silver dressings [56]. For that reason, the search for forms capable of carrying large amounts of active silver ions while maintaining high tissue tolerance are still underway. Scientific research has led to the development of silver in the form of TIAB (titanium-argentum-bezoicum), a complex that contains active silver ions in combination with a carrier—titanium dioxide [121,122,123,124]. The size of ionic silver particles in the TIAB complex is in the range of 7–45 nm, which increases the contact surface of silver with the structures of bacteria and viruses. An additional feature of TIAB is the ability to bind other molecules such as benzalkonium chloride, an active detergent that has bactericidal and keratolytic properties increasing the antibacterial properties of silver microparticles. Currently, the TIAB complex has been used in medical devices in dermatology, gynecology and dentistry, as well as the compound of medications. This complex is gradually becoming more and more popular throughout many European countries [121,122,123,124]. Titanium dioxide (TiO_2_) in the TIAB complex can bind large amount of silver ions in a stable structure, in concentrations higher than those found in popular drugs and medical devices [121,122,123,124]. Besinis et al. (2014) compared the toxic efficacy of various nanomaterials such as AgNP and titanium dioxide against the disinfection of chlorhexidine commonly used against *Streptococcus mutans* strains [121]. Agnihotri et al. (2013) additionally showed that the AgNP mechanism of bactericidal activity, where AgNPs were immobilized with amines on the surface of functionalized silica, showed a greater effectiveness of contact killing than colloidal silver contained in the solution [125]. Silica nanoparticles decorated with silver nanocomposite, the so-called polyrhodanines, show higher potential and effective antibacterial activity against *E. coli*, *S. aureus* and *Candida* species which is due to the special combination of AgNPs and polyrhodanines [9,31,73]. Ionic silver (Ag+) is bound to TiO_2_ by covalent bonds, which creates a construction that Ag+ silver cannot convert to the non-ionic AgO form. Thus TIAB, deprived of non-ionic AgO silver, is responsible for the accumulation in tissues and cells [121,122,123,124,125]. Experimental studies on cell cultures and animals, confirmed the lack of cell toxicity, skin and mucosa irritation caused by TIAB, thus enabling the use of the compound in the treatment of bacterial, fungal and viral infections of, among others, pressure sores, ulcers, burns, mycoses, diabetic foot and skin and mucous membrane infections [121,122,123,124,125]. TIAB complex is effective and safe in treatment of vulvovaginal infections, proctologic diseases (anal fissures, hemorrhoids) and skin diseases [98,121,122,123,124,125]. TIAB in the form of a vaginal gel was used to treat vulvovaginitis, caused by bacterial, fungal and viral infections, mucous membranes and superficial traumas after delivery, as well as to prevent recurrent genital herpes infections (HSV-2) [90,105,111,112]. Clinical examination improvement after TIAB treatment in patients with a clinical diagnosis of mycosis or vulvovaginitis (positive bacterial cultures from *E. coli* isolation, hemolytic streptococci, *P. aeruginosa*, *L. monocytogenes*, *E. faecalis*, *Salmonella enteritidis*, *Candida* spp., *Trichomonas vaginalis*, *Gardnerella*) or in patients with a clinical diagnosis of genital herpes, brought clinical improvement of 82.8% [106,107,108,109,110]. 

### 4.1. Antifungal Properties 

Biologically synthesized AgNP stabilized additionally with sodium dodecyl sulfate and analysis of MIC tests showed increased antifungal activity compared to fluconazole often used in hospitals in relation to clinical isolates and ATCC strains: *Trichophyton mentagrophytes*, *Phoma glomerata*, *Phoma herbarum*, *Fusarium semitectum*, *Trichoderma* sp., *Candida albicans*, *Candida glabrata*, *Aspergillus niger* and *Issatchenkia orientalis* [31,32,33,34,35,36,37,38,39,73,74,75,76]. Biologically synthesized AgNPs showed antifungal activity against phytopathogens such as *Alternaria alternata*, *Sclerotinia sclerotiorum*, *Macrophomina phaseolina*, *Rhizoctonia solani*, *Botrytis cinerea* and *Curvularia lunata Fusarium oxysporum* [31,32,33,34,35,36,37,38,39,73,74,75,76].

The antifungal efficacy of AgNPs was also assessed in combination with antibiotics such as nystatin (NYT) or chlorhexidine (CHX) or chlorhexidine digluconate (CHG) against biofilms formed by *C. albicans* and *C. glabrata*, *Bipolarisum sorokiniana, Penompicillium sorokiniana, Penompicillium sorokiniana, Aspergillus fumigatus, Cladosporium cladosporoides, Chaetomium globosum, Stachybotrys chartarum* and *Mortierella alpine* grown on agar [31,32,33,34,35,36,37,38,39,73,74,75,76].

### 4.2. Nanoparticles in Angiogenesis

Angiogenesis, as a process of forming new blood vessels, takes place either under physiological (e.g., embryonic development) or pathophysiological (e.g., growth of cancerous tumors) conditions. The data show that carbon nanoparticles are anti-angiogenic [89,126].

In chicken embryo studies, it has been shown that diamond and graphite nanoparticles reduce the weight of the heart and the network of its blood vessels without affecting the mass of other organs and the entire body. In addition, diamond and graphite nanoparticles inhibit the expression of factors activating VEGF and FGF2 angiogenesis at the level of mRNA and protein in the heart of a chicken embryo after the first day of incubation of eggs into the air chamber [34,35,92]. The use of metal nanoparticles (Ag, Au) increased the expression of mRNA and angiogenesis inducing protein—VEGF, cell proliferation—FGF [90], myogenesis by MYOD1 and ATP synthesis (ATPase Na+/K+ transporting subunit alpha 1 (AT-P1A1). Using carbon nanoparticles, especially diamond ones, showed rather opposite trends [35].

Therefore, nanoparticles can be applied in medicine to normalize processes related to the development of blood vessels. According to the needs, nanoparticles of a pro- or anti-angiogenic nature can be used. AgNPs as hydroxyproline carriers, administered to the egg on day 1 of chicken embryo incubation, increased the thickness of collagen fibers in blood vessels and the development of new vessels, which may be used in stimulating the development of the chicken embryo’s circulatory system and in human cardiovascular diseases [34,35,92]. Silver nanoparticles can also be an effective carrier for amino acids or ATP. When administered by the in ovo method, they stimulated the development and maturation of muscle fibers by activating PCNA and increasing the surface area of breast muscle fibers. The observed mechanism was associated with an increase in the metabolic rate determined by measuring oxygen uptake (Figure 6) [34,35,92]. Pathological angiogenesis is symbolic of cancer and various ischemic and inflammatory conditions [89,126]. Kalishwaralal et al. (2009) demonstrated the anti-angiogenic properties of biologically synthesized AgNP using bovine retinal endothelial cells (BREC) as a model where they found inhibition of proliferation and migration in BREC after 24 h of treatment with AgNPs at a 500 nM concentration [92]. Induced mechanisms of VEGF inhibition angiogenic process through caspase-3 activation and DNA fragmentation, and AgNP inhibited the VEGF-induced PI3K/Akt pathway in BREC in the formation of new blood microvessels [92].

## 5. Negative Effects of Silver Nanoparticles

The negative mechanisms of action of the silver particles are reported for colloidal silver. Colloidal silver (Argentum colloidale), also known as collargol, is a pharmaceutical raw material for the production of prescription drugs, which is a combination of silver with protein or gelatin. Colloidal silver in the particle size range from 1 to 100 nm can exert neurotoxic effect on people. It weakens fertility and accumulates in liver, pancreas and lungs, damaging a microbiome entirely, even more than antibiotics [61,101]. The mechanism of colloidal silver action was described as follows: The colloidal silver carries oxygen to atomic oxygen, which reacts with thiol groups of cysteine, catalyzes the formation of sulphide bonds, and surrounds the cells by sticking on the flagellum and preventing their movement [65]. It leads to preventing the synthesis of cell walls during cell division and the death of bacteria. Other cellular structures, necessary for their functioning such as polarized cell membrane, are also sensitive to nanoparticles. Colloidal nanosilver interferes with an action of the sodium-potassium pump, affecting the transport of nutrients to the interior of bacteria. In addition, the accumulation of silver particles in the cell membrane leads to an increase in permeability, disruption of cellular ions and loss of intracellular ATP [36,37,50,99]. Nanosilver generates free radicals (protons) production causing the destruction of disulphide bonds, disruption of tertiary structures and ultimately protein denaturation [36,37,50,99]. However, the harmful effect of bacteria is completely inhibited by enzyme inactivation. Silver combined with the −SH groups blocks the initiation of metabolic reactions. An abnormal action of cytochrome oxidase or succinate dehydrogenase will inhibit the transfer of oxygen molecules from cytochromes c to mitochondria and prevent the process of oxygen molecules reduction. Silver also has a detrimental effect on amino acids [127,128,129,130,131,132,133,134,135,136,137,138,139,140]. By combining with the groups NH_2_, COOH and C_3_H_4_N_2_, it denatures essential amino acids in many important metabolic pathways (tyrosine dephosphorylation) and leads to the loss of biological activity of the pathogen. It is also believed that silver penetrating cells of microorganisms also reacts with nucleic acids and inhibits the DNA replication process in different types of normal and cancer in Eucaryotic cells [97,111,112,127,128,129,130,131,132,133,134,135,136,137,138,139,140]. 

Chemical synthesis requires the use of many compounds (often toxic), which are reducers or stabilizers that affect the toxicity of the silver nanoparticles manufacturing process and are a potential threat to the natural environment [141,142]. One of the most challenging problems in modern organic chemistry and technology is mapping what happens to disintegrated plastic in the natural environment. There is a considerable risk that plastic waste releases nano-sized particles known as nanoplastics due to their potential impact on the environment. People use a lot of plastic in their life to preserve food, keep warm, communicate with the world, etc., which exerts a big harm on the life and environment. The use of AgNPs is growing rapidly: food industry, construction, medicine, personal items, washing machines, cosmetics, wall paints, water disinfectants, air fresheners, guides, mirrors, photography. Since the consumption of those products in the population is higher, AgNPs accumulate in kidneys, liver, intestines, tongue and brain. They cause cell death, through DNA damage. A production of free radicals cause inflammation penetrating the blood–brain barrier. When administered to female mice by i.p., accumulation in fetal nervous system were observed (Figure 7) [98,137]. 

Nanoparticles of various metals, such as silver, gold, copper, titanium oxides, zinc and iron, can enter the body in three different ways: by inhalation, by ingestion while eating or drinking and through the skin. The most popular NPs metal is titanium dioxide [29,121,123,125]. As opposed to silver ions, its nanoparticles show a strong tendency to aggregation, which makes them lose their positive properties. On the other hand, silver ions, due to the presence of charge, have a high affinity to functional groups that are also part of human cells [29,93]. Therefore, work is still ongoing to create particles with strong antibacterial properties that are safe for eukaryotic cells. These particles would not aggregate, leading to a loss of physicochemical properties and, consequently, a reduction of biological activity [143,144,145,146,147,148,149,150,151]. The way to solve this problem and maintain high efficiency of nanopreparations was the immobilization of AgNPs on carriers, such as titanium dioxide [119,123,124]. By sharing electrons, characteristic of covalent bonds, only the silver ionic forms (Ag+) are present in the complex [121,152,153,154,155,156,157,158,159]. The tests show a much greater range of action of the TIAB complex than the commonly used chlorhexidine. This effectiveness was confirmed by determining the MIC value, which in the case of silver and its complexes was much lower than that of chlorhexidine [121]. Titanium NPs are commonly found as a food additive E171 often used in sweets and chewing gum. Studies in mice have shown that it penetrates the blood–brain barrier and is neurotoxic, and by accumulating in the nervous system of mice fetuses, it impairs their development [152]. Therefore, this type of nanoparticles is being abandoned and replaced with silver nanoparticles, which have gained enormous popularity in the recent years. Silver NPs are also present in cosmetics, paints, deodorants, antibacterial clothing (socks, sportswear), disinfectants, washing machines, refrigerators, air fresheners, tattoo inks, dressings, self-cleaning paints or even mirrors. However, apart from this positive side, silver also has a negative one. Silver nanoparticles have the unique ability to accumulate in the internal organs of various animals. The distribution of nanoparticles in different organs tested in animals may vary over time. Studies in rats exposed for 90 days by the oral route to AgNPs with a size of 60 nm showed they accumulated in female kidneys, bladder and adrenal glands (Figure 7) [121,153,154,155,156,157,158,159]. Comparative studies of the effects of nanoparticles and silver ions show that silver ions are absorbed faster and cause more changes in the body than its nanoparticles. Silver nanoparticles (14 nm) along with silver acetate were administered to male rats by gavage for 28 days. The distribution of both forms of silver (nanoparticles and ions) was similar. However, the highest amounts of silver outside the intestines were found in kidneys and liver and lower amounts in lungs and brain [121,153]. After intravenous injection of 20 nm AgNPs in Wistar rats at a dose of 5 mg/kg BW and collecting biological samples from them after 24 h, 7 or 28 days, the highest concentration of nanoparticles in the samples was recorded in the liver collected after 24 h after the administration. After 7 days, high levels of silver were observed in lungs and spleen, while in kidneys and brain, silver levels peaked 28 days after injection [121,154]. Once in the bloodstream, nanoparticles can accumulate in various organs. In rats dosed intravenously with AgNPs ranging in size from 20 or 80 nm for 5 consecutive days, nanoparticles were detected in lungs, liver and spleen [121,155]. In a similar experiment carried out on mice, which received nanoparticles with a size of 12 nm for 24 h after exposure, the presence of silver in spleen and liver has also been demonstrated (Figure 7), [121,154,155,156,157,158,159]. Studies on the absorption of AgNPs were also conducted on skin of guinea pigs. After a 14-day application of the AgNPs suspension (size of 20 to 50 nm) on pigs’ skin, nanoparticles were detected only in the stratum corneum (Figure 7), [121,154,155,156,157,158,159]. Conjunctival hyperemia was observed, which subsided after 72 h after the administration of colloidal silver (size of 10–20 nm) into the conjunctival sac, which indicated a temporary irritating effect [121,152,159]. Silver nanoparticles, in addition to their wide application in inhibiting the growth of bacteria, fungi and viruses, also play an important role in the human body. People are exposed to silver compounds daily, mainly by breathing, drinking water, eating food, containing AgNPs. Scientists estimate that 98–99% of the supplied silver in the form of compounds is excreted from the body on the second day, and the remaining 1–2% is accumulated in the human body. Prolonged exposure of human skin to silver compounds such as silver nitrate, silver acetate and argyrol in high doses and with physicochemical properties specific to the compound with exposure to sunlight can lead to a blue discoloration of the skin called argyria. This is a disease caused by excessive accumulation of silver in the body. It can be caused by the accumulation of AgNPs in the dermis (when administered topically) or by stimulation of melanin to produce silver periwinkle (when administered orally or intravenously), [29,62,74,121]. Symptoms of the disease appear after about 6 months (or even a year—depending on the dose taken) of regular silver intake. The first characteristic symptom of silver disease is blue discoloration of the gum line. Other symptoms include irreversible changes in the color of the skin (selected areas or the entire surface) and nails. If the patient additionally regularly uses eye drops containing silver compounds, the color of the eyeballs may change as a result of the accumulation of the element in the cornea and capsule of the anterior lens. Subsequently, an excess of silver compounds in the body can cause kidney disease, liver disease (the color of internal organs blue or bluish-gray) and hardening of the arteries, and the deposition of silver in the eye sockets can impair vision. The lowest dose that can cause symptoms of poisoning is 0.014 mg of silver per kilogram of body weight per day, administered intravenously. In extreme cases, silver can be found in moderate amounts in urine and feces, as well as in saliva [29,62,74,121]. 

## 6. Other Properties of Silver Nanoparticles

Silver nanoparticles are also used in in vitro and in vivo bioengineering to create corneal implants (replacements) or permeable contact lenses on collagen matrices of various colors—taking advantage of their physical and physicochemical properties [143,144]. Silver nanoparticles can be incorporated into acrylic resins to make removable dentures for prosthetic treatment, composite resins for restorative treatment, irrigation solutions and fillers for endodontic treatment, adhesive materials for orthodontic treatment, guided membranes for tissue regeneration for periodontal treatment and titanium coatings in implantology treatment. A common problem, however, is the potential hazard if released into the environment. The interaction of nanoparticles with toxic materials and organic compounds may reduce their therapeutic properties [68,145,146]. 

Theranostic applications as the future of personalized precision nanomedicine relate to diagnosis and therapy demonstrated in Astilean [145], in which SERS imaging and photothermal therapy are coupled together. Currently, the teranostics uses innovative, newest technologies and materials, such as nanotechnologies, nanomaterials, biomaterials and biomimetics, which simultaneously allow for the detection of disease processes in the patient’s body and for taking actions that enable effective delivery of drugs directly to the areas affected by the disease process, which allows for more effective treatment to the individual needs of a particular patient.

An example of the theranostic use of AgNPs is their application in the treatment of triple negative breast cancer (TNBC). The disease is associated with the worst clinical outcomes due to the aggressive nature of the tumor, delayed diagnosis, and non-specific symptoms in the early stages. Therefore, the identification of new specific TNBC serum biomarkers for screening and therapeutic purposes remains an urgent clinical need. Silver nanoparticles dispersed in the blood are covered with a special protein coating, the so-called “Protein crown” (PC). Traditional changes in protein patterns are difficult to detect with conventional blood analysis, PC acts as a “nanoconcentrator” of serum proteins with affinity for the surface of AgNP. Thus, characterization of the PC could allow the detection of otherwise undetectable changes in protein concentration at an early stage of the disease or after chemotherapy or breast surgery using liquid chromatography methods, tandem mass spectrometry (LC-MS/MS) with confirmation through a sequence window acquisition of all theoretical mass spectra (SWATH) with an AgNP size of approximately 10 nm. These AgNPs, interacting with human blood serum, can also be helpful in diagnosing other diseases defined as civilization [94,95,126,127].

Other properties of silver nanoparticles are their use in dentistry and periodontology as fillings in tooth fillings and the creation of a permanent scaffold in the tooth defect.

The growing number of infections caused by microorganisms in devices commonly used in medical operations or dental procedures, e.g., for surgery or implantation of bone defects in teeth, encourages the search for new biocompatible surfaces with stronger and more effective antibacterial and antiviral properties. The attention of scientists was drawn to silver nanoprisms, which can be used to cover various types of surfaces, enriching them with new, unique antibacterial and antiviral properties [147,148,149,150,151]. In order to inhibit the adhesion and proliferation of bacteria and viruses on biomaterials, several strategies of surface functionalization were applied, such as bactericidal titanium surfaces using poly-L-lysine (PLL) as a mediator after sputtering AgNP on surfaces [148]. The antibacterial activity of that surface was tested against two different bacteria strains *S. aureus* and *P. aureginosa* (Figure 8). For both strains and different sizes of AgNP, the surface modified with PLL and AgNPs shows significantly enhanced antimicrobial activity compared to titanium deposited AgNPs by 5 rows relative to the original inoculum bacteria. That means that the relatively low AgNPs load on PLL-modified titanium surfaces reaches 99.999% of bacterial death after 24 h. 

A new approach to the use of silver nanoparticles was their use in skin and epithelial diseases. Since diseases of the skin and epithelium occurring in intimate places are still a difficult topic, new solutions and methods of wound treatment are being intensively searched for, in particular, the creation of an ideal dressing. A dressing intended for the treatment of chronic wounds, such as burns and diabetic wounds, should be effective against exudates and provide a microenvironment conducive to repair processes. In the case of critically colonized wounds, the dressing should penetrate the biofilm and reduce pathogens. Tests carried out on AgNPs confirm the possibility of penetration of particles into the biofilm layer [38,62,74,78,80,81], which is extremely important when fighting the source of infection or colonized spaces [97,111,112,152,153,154,155,156,157,158,159,160,161,162]. There are many products on the market today, including nanosilver dressings. Modern dressings combine silver nanoparticles with substances destabilizing the biofilm matrix, which significantly increases their antibacterial effectiveness. Dressings should be characterized by absorbent and autolytic properties for topical application, where the healing process has stopped in the inflammatory phase. Such dressings with AgNPs should be highly effective: causing a significant reduction of the biofilm mass and the titer of pathogenic bacteria in infected wounds. The influence of pre-treatment of cotton fabric with biopolymer chitosan (CHT) on the deposition of colloidal triangular AgNPs was investigated. The influence of deposited silver nanoparticles on the color and antibacterial activity of cotton fabrics was also assessed. Characterization of colloidal silver nanoparticles and AgNPs deposited on cotton fabrics was carried out using electron microscopy (TEM and FESEM), XRD analysis, atomic absorption spectroscopy, UV–VIS absorption and reflection spectroscopy (Figure 8). The cotton fabric turned from white to blue after deposition of triangular silver nanoplates. The antimicrobial activity of the CHT pre-impregnated cotton fabric impregnated with AgNPs was tested on *E. coli* gram-negative bacteria, *S. aureus* gram-positive bacteria and *C. albicans* fungi [9,30,32,43,44,46,54,73,106,107,108,109,110]. The embedded AgNPs gave the cotton fabric excellent antibacterial properties. The cotton fabric standard sterilization procedure for testing antimicrobial activity caused the fabric color to change from blue to yellow. This color change is most likely a consequence of the transformation of triangular silver nanoplates into nanodisks and/or their agglomeration into spheroids (Figure 8). Another very important advantage of nanoparticles is the high surface to volume ratio, i.e., the larger diameter increases the chemical activity of nanoparticles [163]. The optical properties also depend on the particle size. For example, a red solution of colloidal gold turns yellow as the particle size increases [164]. The highly developed specific surface area has a significant influence on the adsorption properties, reactivity of materials and other properties, including the described antimicrobial activity [165,166].

## 7. Conclusions

This review comprehensively covers the issues related to microbiological and medical applications of silver nanoparticles in diseases caused by pathogens of bacteria, viruses or fungi, with particular emphasis on angiogenic, anticancer and therapeutic effects as a component of many drugs in clinical trials and in antibiotic therapy. The growing drug resistance of bacteria, especially those developing in biofilms, to conventional antibiotics is triggering an intensive search for new therapeutic agents. Due to the proven antibacterial potential, AgNPs are intensively studied. The mechanism of their antibacterial action is mainly based on the damage to the bacterial envelope and the induction of reactive oxygen species. Silver nanoparticles, in addition to their high antibacterial potential, are able to interact with conventional antibiotics, significantly increasing their antibacterial properties, as well as characteristics of potential methods of its synthesis, and other additional biological applications targeted at individual cell components. The treatment of periodontal diseases and severe periodontitis with an active additive of AgNPs as well as AgNPs as ingredients of many toothpastes or special dental and prosthetic fillings deserve attention.

What can be a new alternative in civilization diseases such as atherosclerosis; osteoporosis; type 1 and 2 diabetes and neurodegenerative diseases such as Parkinson, Alzheimer or Wilson’s disease? What will ensure the patient safe treatment of these disease entities in a non-invasive manner? This publication is much more than a compendium of knowledge for doctors, microbiologists and scientists dealing with such issues. All answers to the questions about AgNPs bothering us are included in this review. Veterinarians will find it also advantageous because we described positive or negative effects of AgNPs on animal models to prevent potential human diseases, called zoonoses. The wide application of nanoparticles makes it necessary to closely monitor their synthesis and use also in terms of the negative effects of their operation.

## Figures and Tables

**Figure 1 ijms-22-00854-f001:**
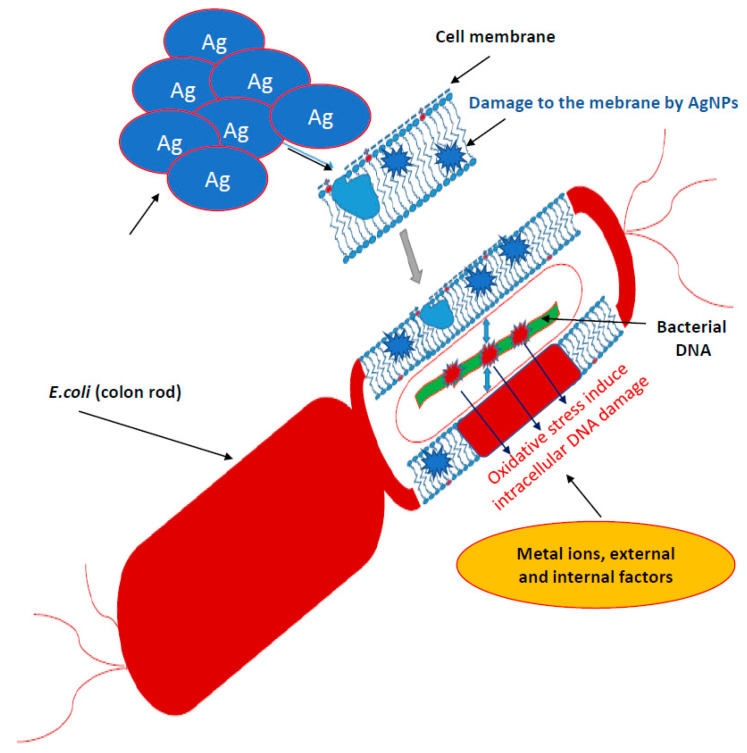
Mechanisms of silver nanoparticles action into bacterial cells. It is characterized by a lack of specificity and runs at a very high speed, causing oxidative stress that induces bacterial DNA damage.

**Figure 2 ijms-22-00854-f002:**
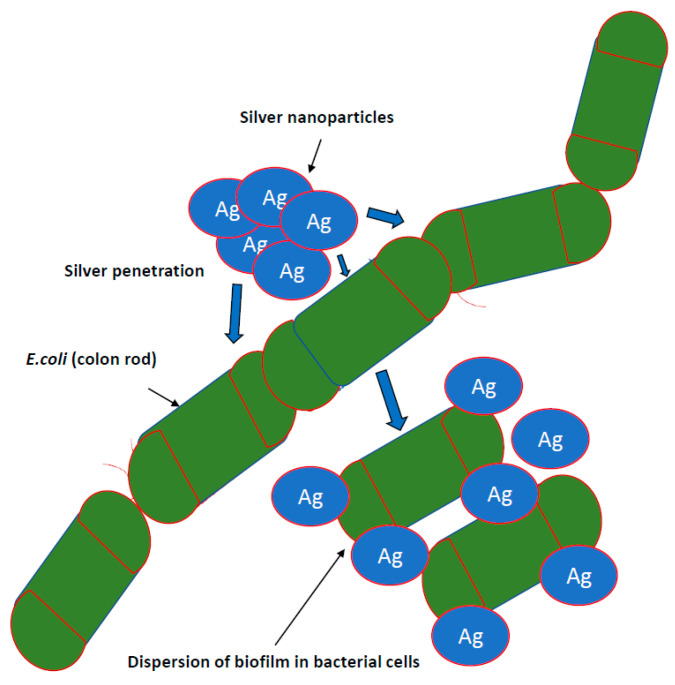
Inhibiting the growth of bacterial biofilms by nanoparticles. It is known that nanosilver efficiently prevents the formation and further stages of biofilm development; however, the effect depends on the species and even the bacterial strain. This effect is generally stronger against biofilm formed by Gram-negative pathogenic bacteria.

**Figure 3 ijms-22-00854-f003:**
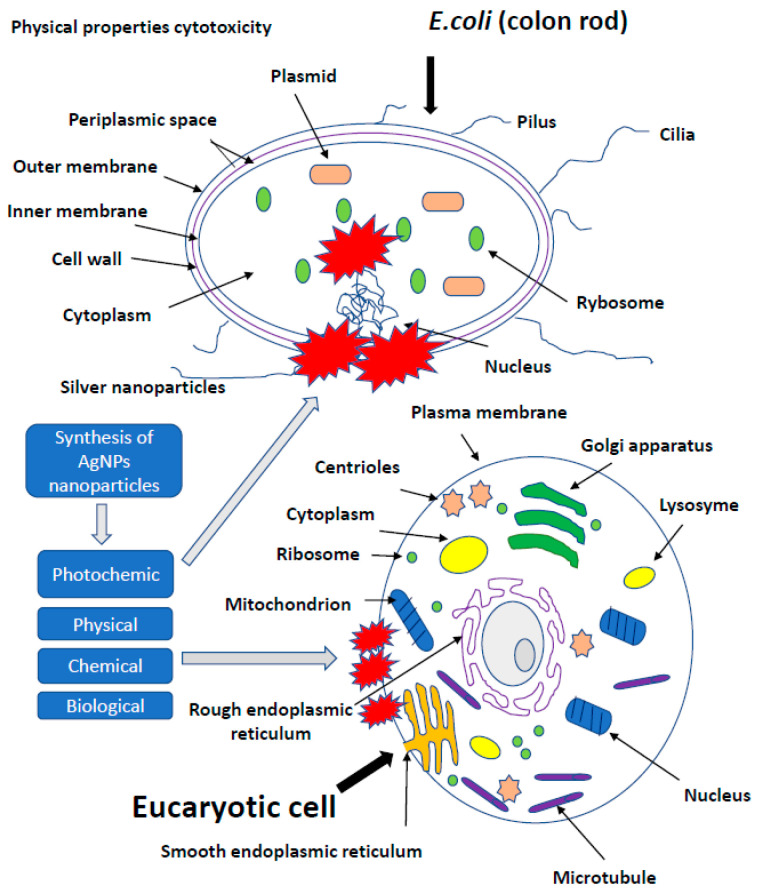
The effect of silver on more developed organism. It is noted that clinical symptoms, such as wound exudate, lower the cytotoxicity of AgNPs, and in this case, the high concentration of proteins in the exudate may neutralize the toxicity of nanosilver to neighbouring tissues.

**Figure 4 ijms-22-00854-f004:**
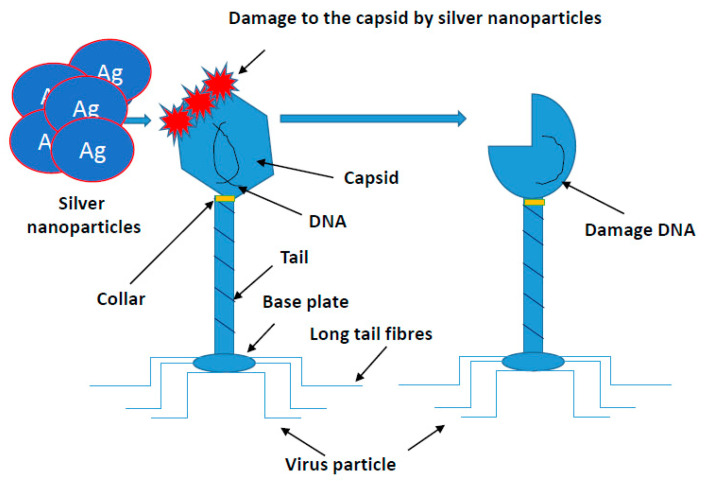
Mechanism of silver nanoparticles action on viruses’ structure. All viruses replicate through a similar sequence of events. AgNPs-based viral vaccines can be very effective against HIV-1, malaria, hepatitis B syncytial respiratory virus, HSV-1, monkey pox virus, influenza virus and Tacaribe virus.

**Figure 5 ijms-22-00854-f005:**
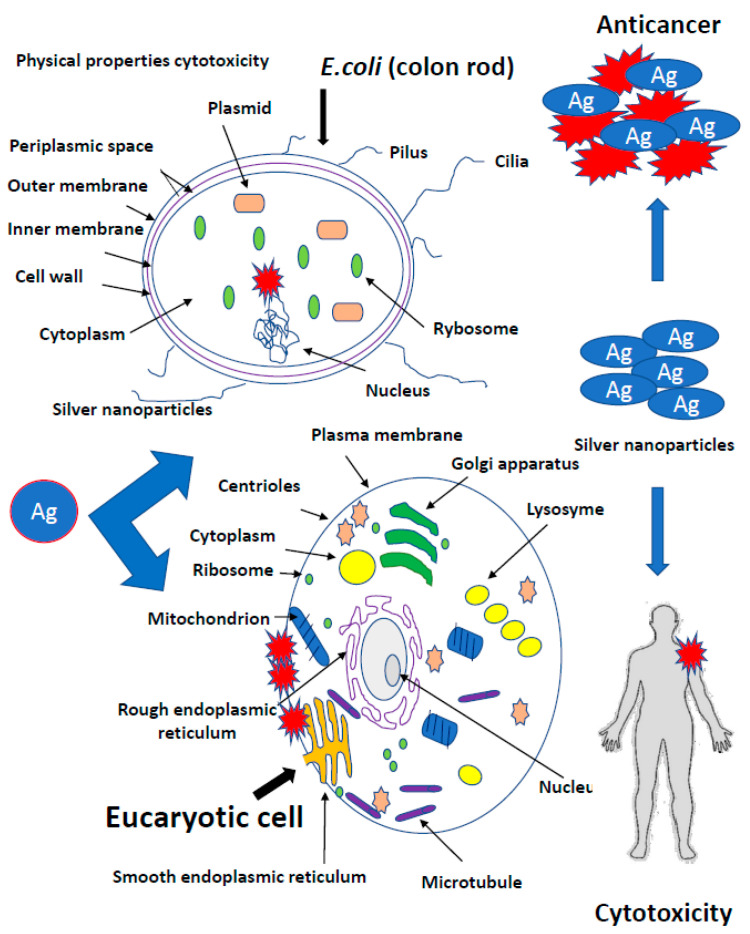
Toxicity of silver nanoparticles in different type of tissues. The role of toxicokinetic and toxicology of nanomaterials (especially in oral contact) and their effect on the environment.

**Figure 6 ijms-22-00854-f006:**
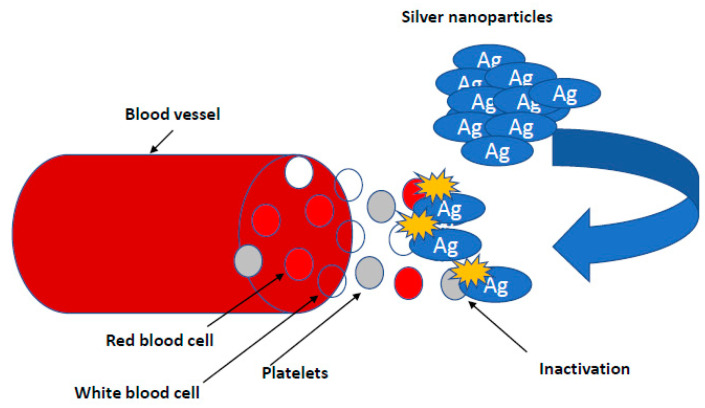
Nanoparticles in research on angiogenesis. Silver nanoparticles can also be an effective carrier for amino acids or ATP. When administered by the in ovo method, they stimulated the development and maturation of muscle fibers by activating PCNA and increasing the surface area of breast muscle fibers. The observed mechanism was associated with an increase in the metabolic rate determined by measuring oxygen.

**Figure 7 ijms-22-00854-f007:**
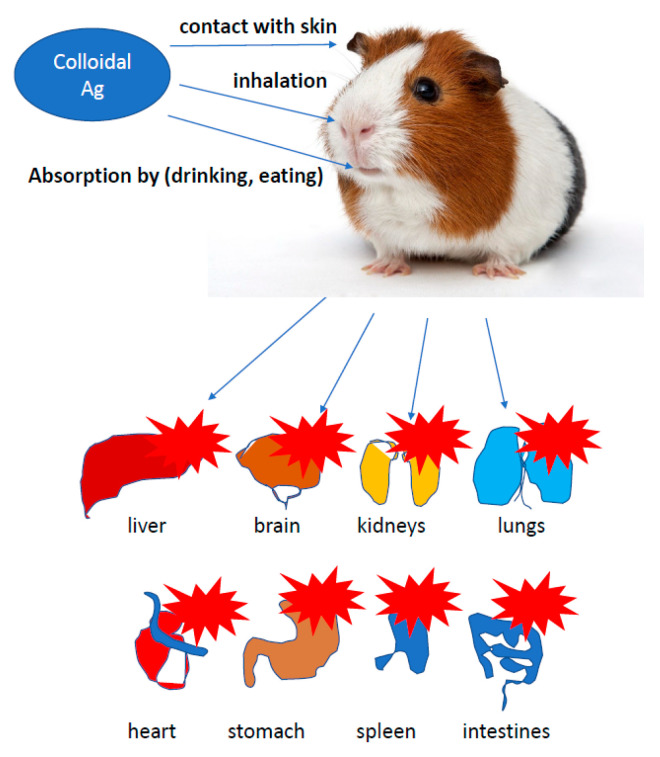
Negative role of colloidal silver. Colloidal silver can accumulate in different tissues of rodents like kidneys, liver, intestines, tongue and brain. They cause cell death, through DNA damage.

**Figure 8 ijms-22-00854-f008:**
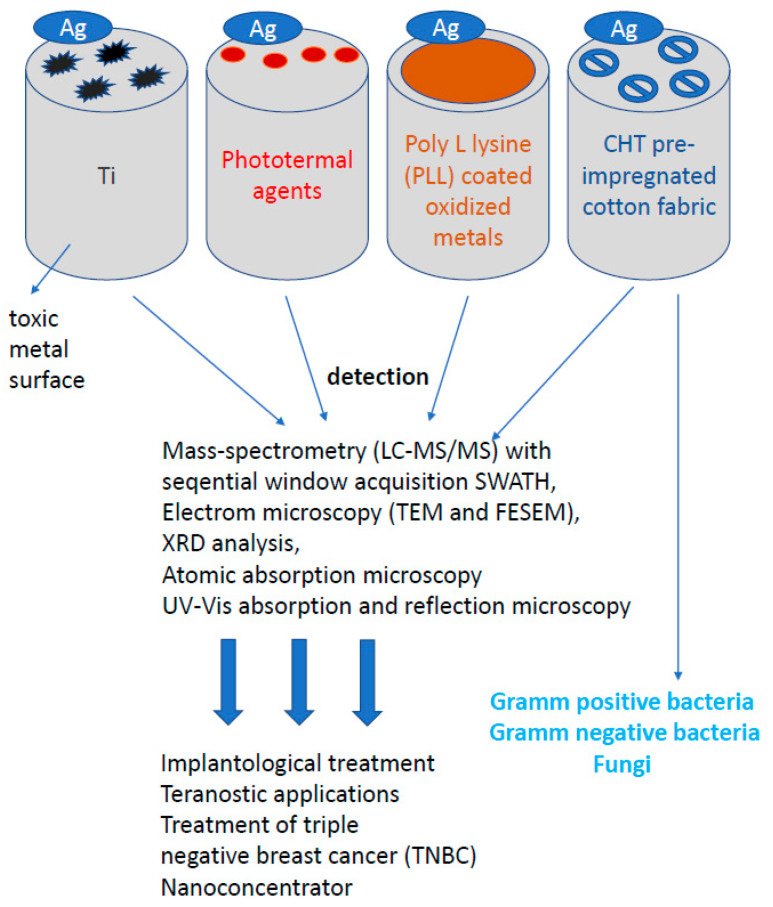
Other properties of silver nanoparticles. Despite the wide anisotropic and anti-inflammatory properties of silver nanoparticles (AgNP) in the human body, silver, in addition to its widespread use on bacterial or fungal cells, also plays an extremely important role in bioengineering.

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
