# Peer review of "All That Glitters Is Not Silver—A New Look at Microbiological and Medical Applications of Silver Nanoparticles"

_ijms, 2021, doi:10.3390/ijms22020854_

Round 1

Reviewer 1 Report

The author has made a very detailed summary of the research status and application of silver nanoparticles. However, it is recommended that the author carefully modify all the figures in their manuscript, because the layout and interpretation of the existing figures are confusing and can easily mislead readers. So the reviewer suggested a major revision before publication.

Author Response

Rebuttal Letter

Dear Editor,

We are grateful for the review of our manuscript ‘All that glitters is not silver – a new look at microbiological and medical applications of silver nanoparticles’ (IJMS-1053577). We have incorporated the suggestion as proposed by the Reviewers. The exact changes made in the main text are typed in red. The specific response to the Reviewers’ comments is given below.

Reviewer 1:

Comment: The author has made a very detailed summary of the research status and application of silver nanoparticles. However, it is recommended that the author carefully modify all the figures in their manuscript, because the layout and interpretation of the existing figures are confusing and can easily mislead readers. So the reviewer suggested a major revision before publication.

Response:  We truly appreciate that the Reviewer had paid attention to the figures. We have made some changes, but it would have been easier for us to make corrections if the Reviewer had specified his preferences for the appearance of the figures more precisely. We greatly appreciate all critical comments.

Sincerely,

Pawel Kowalczyk

Reviewer 2 Report

The review "All that glitters is not silver - a new look at microbiological and medical applications of silver nanoparticles".

The review article is a good introduction to the topic of silver nanoparticles. Proposes their use as antibacterial agents along with the mechanisms of action, as well as their action on more developed organisms - anti-cancer, antiviral, antifungal properties. An important part is the chapter on the negative interactions of AgNPs. Do the authors see a similarity with the departure from the use of titanium oxide nanoparticles? Is such a future for AgNPs?

While the article is interesting, it is one of many reviews on AgNPs. Each year there is at least one review on this topic.

I would expect more of the current literature - 2019 and 2020. From 2020, the authors cite only 5 articles and from 2019 only 6. Among the citations, you can find 5 citations of the team that created the review (with 2 citations from 2020) of the year. I wonder why the authors did not base their review on current discoveries and achievements. Citations from, for example, 2005, 2008, 2009 or 2014 do not constitute the novelty of this work.

Please present more current literature in this review article (2019-2020). It is not possible that in the subject in which the authors specialize, there were only 11 works from the mentioned years (out of 141 cited).

Author Response

Rebuttal Letter

Dear Editor,

We are grateful for the review of our manuscript ‘All that glitters is not silver – a new look at microbiological and medical applications of silver nanoparticles’ (IJMS-1053577). We have incorporated the suggestion as proposed by the Reviewers. The exact changes made in the main text are typed in red. The specific response to the Reviewers’ comments is given below.

Reviewer 2:

Comment: The review article is a good introduction to the topic of silver nanoparticles. Proposes their use as antibacterial agents along with the mechanisms of action, as well as their action on more developed organisms - anti-cancer, antiviral, antifungal properties. An important part is the chapter on the negative interactions of AgNPs. Do the authors see a similarity with the departure from the use of titanium oxide nanoparticles? Is such a future for AgNPs?

While the article is interesting, it is one of many reviews on AgNPs. Each year there is at least one review on this topic.

I would expect more of the current literature - 2019 and 2020. From 2020, the authors cite only 5 articles and from 2019 only 6. Among the citations, you can find 5 citations of the team that created the review (with 2 citations from 2020) of the year. I wonder why the authors did not base their review on current discoveries and achievements. Citations from, for example, 2005, 2008, 2009 or 2014 do not constitute the novelty of this work.

Please present more current literature in this review article (2019-2020). It is not possible that in the subject in which the authors specialize, there were only 11 works from the mentioned years (out of 141 cited).

Response: We are very grateful to the Reviewer for this valuable consideration. Yes, we observe the tendency that the initially used titanium nanoparticles were discontinued after careful study and interactions on human health. A similar process can be observed in the case of silver nanoparticles, but on a smaller scale as silver is now widely used in all industries, especially in medicine, including dentistry and micorbiology. However, silver has a dark side, similar to molecular oxygen. It can also cause inflammation, or allergic reactions to human skin. In our review paper, we also wanted to draw attention to this by showing the negative effect of silver nanoparticles.

As suggested by the reviewer, we updated the literature from 2019 and 2020, in total there are added 50 items from both years, which is 1/3 of the entire article. It is known that some achievements regarding silver nanoparticles were discovered earlier, hence a handful of articles from earlier years. However, we tried to make the topics we describe as up-to-date and modern as possible.

All amendments concerning the updated literature are typed red.

Sincerely,

Pawel Kowalczyk

Reviewer 3 Report

The review has an interesting approach, but, as the review on Antibacterial AgNP are really a lot, authors should try a further step in describing the landscape related to the topic.

In general, i see small attention, in this review, for anisotropical silver nano-objects, which have shown quite interesting antibacterial applications. see for example J. Mater. Sci. 2014, 49, 4453–4460 and nanoscale 2016, 8, 6484-6489, just to make a couple of examples. Silver nanoprism has shown to have a lot of pontential in antibacterial application coupled to variability of colours that they can impart to surfaces

there is also another quite big absence, connected to the previous: the use of anisotropic AgNP as photothermal agents, a feature which can be used in cooperation with classical antibacterial action of silver NP or for other puroposes...as authors state about a new look, they should include some unconventional  example of silver NP use, as in the papers from Astilean and co-workers, see for example  Mol. Pharm. 2014, 11, 391–399 and Cancer Lett. 2011, 311, 131–140  in which authors report the use of photothermal silver nanoparticles for theranostic applications

For the antibacterial use of photothermal silver nanoparticles, see for example RSC advances 2016 6 (74), 70414-70423, Nanomaterials 2017  7 (1), 7, where the antibacterial silver nanoprisms are grafetd on surfaces.... and here we go for another part that in my opinion should be completed

In fact, few word and citation are spent for bulk antibacterial surfaces based on AgNP. see some papers from Schilardi and collegues: ACS Appl. Mater. Interfaces 2018, 10, 28, 23657–23666, ACS applied materials & interfaces 5 (8), 3149-3159; and from Pallavicini:  Scientific reports 2017 7 (1), 1-10, European Journal of Inorganic Chemistry 2018 (45), 4846-4855

these gaps should be in my opinion filled for a compherensive review which wants to give a new look on AgNPs. the papers proposed here are just a starting point, maybe authors could try to find some more, related to the specific topics this referee has suggested.

other points

authors state that

"Silver in the form of nanoparticles melts at 360°C, while silver metal at 960°C."

are authors sure that all AgNP melt at 360 °C? I suppose it depends on size (and maybe shape...). can authors add a reference to justify the affirmation?

for the virus part and the covid emergency, Molecules 2011, 16, 8894-8918 should be cited for potential of agnp as antivirus agent, and more specificall a recent review on covid 19 and nanotech should be integrated and cited. Nanomaterials 2020, 10, 802

Author Response

Rebuttal Letter

Dear Editor,

We are grateful for the review of our manuscript ‘All that glitters is not silver – a new look at microbiological and medical applications of silver nanoparticles’ (IJMS-1053577). We have incorporated the suggestion as proposed by the Reviewers. The exact changes made in the main text are typed in red. The specific response to the Reviewers’ comments is given below.

Reviewer 3:

Comment: The review has an interesting approach, but, as the review on Antibacterial AgNP are really a lot, authors should try a further step in describing the landscape related to the topic.

In general, I see small attention, in this review, for anisotropical silver nano-objects, which have shown quite interesting antibacterial applications. See for example J. Mater. Sci. 2014, 49, 4453–4460 and nanoscale 2016, 8, 6484-6489, just to make a couple of examples. Silver nanoprism has shown to have a lot of pontential in antibacterial application coupled to variability of colours that they can impart to surfaces.

There is also another quite big absence, connected to the previous: the use of anisotropic AgNP as photothermal agents, a feature which can be used in cooperation with classical antibacterial action of silver NP or for other puroposes...as authors state about a new look, they should include some unconventional  example of silver NP use, as in the papers from Astilean and co-workers, see for example  Mol. Pharm. 2014, 11, 391–399 and Cancer Lett. 2011, 311, 131–140  in which authors report the use of photothermal silver nanoparticles for theranostic applications.

For the antibacterial use of photothermal silver nanoparticles, see for example RSC advances 2016 6 (74), 70414-70423, Nanomaterials 2017  7 (1), 7, where the antibacterial silver nanoprisms are grafetd on surfaces.... and here we go for another part that in my opinion should be completed

In fact, few word and citation are spent for bulk antibacterial surfaces based on AgNP. see some papers from Schilardi and collegues: ACS Appl. Mater. Interfaces 2018, 10, 28, 23657–23666, ACS applied materials & interfaces 5 (8), 3149-3159; and from Pallavicini:  Scientific reports 2017 7 (1), 1-10, European Journal of Inorganic Chemistry 2018 (45), 4846-4855

these gaps should be in my opinion filled for a compherensive review which wants to give a new look on AgNPs. the papers proposed here are just a starting point, maybe authors could try to find some more, related to the specific topics this referee has suggested.

other points

authors state that

"Silver in the form of nanoparticles melts at 360°C, while silver metal at 960°C."

Response: In spite of the recent report of the in situ high-resolution transmission electron microscopy observation of the melting of Ag nanoparticles (NPs) in Chen et al. [Appl. Phys. Lett. 96, 253104 (2010)], no consistent experimental investigation coupled with theoretical analysis has been reported so far. We report the size-dependence of the melting temperature (Tm) of Ag NPs by both differential scanning calorimetry experiments and thermodynamic assessments. Thermodynamically calculated Tm for the Ag NPs showed a nonlinear function with respect to the inverse of the particle size and agreed well with the present and reported experimental results within an error of 1 K.

Comment: “are authors sure that all AgNP melt at 360 °C? I suppose it depends on size (and maybe shape...). can authors add a reference to justify the affirmation?”

for the virus part and the covid emergency, Molecules 2011, 16, 8894-8918 should be cited for potential of agnp as antivirus agent, and more specificall a recent review on covid 19 and nanotech should be integrated and cited. Nanomaterials 2020, 10, 802

Response: We are grateful to the reviewer for this comment. All amendments concerning the updated literature are typed red in the manuscript.

Once again thank you for your time to review our manuscript. I am very grateful for all comments and recommendations. I am open to further suggestions and look forward to hearing from you.

Sincerely,

Pawel Kowalczyk

Round 2

Reviewer 1 Report

This work can be accepted in present form.

Author Response

Thank you very much for accepting the work in its current form.

Reviewer 2 Report

Thank you very much for all the explanations and following my guidelines when re-editing and rebuilding the article, in particular for adding the latest literature.

I am concerned about the addition in lines 115/116 K degrees ("Silver in the form of nanoparticles melts at 360°K, while silver metal at 960°K.") How is this possible, since ordinary silver melts at 960°C ?! Please clarify and rectify.

The rest of the article is written correctly and provides a current overview about microbiological and medical aplications of AgNPs.

Author Response

Comment: Thank you very much for all the explanations and following my guidelines when re-editing and rebuilding the article, in particular for adding the latest literature.

I am concerned about the addition in lines 115/116 K degrees ("Silver in the form of nanoparticles melts at 360°K, while silver metal at 960°K.") How is this      possible, since ordinary silver melts at 960°C?! Please clarify and rectify.

Response:  We are very grateful to the Reviewer for this valuable consideration. A detailed description of melting points with references is given in lines 115 to 119.

“Silver nanoparticles with a diameter of 2.4 nm melt at an approximate temperature 360°C, while unground silver without additives melts at 961.9 °C, however, boils at 2210°C [17, 18, 19]. In addition, its physical properties such as thermal, electrical and magnetic conductivity, also vary with the size of the silver nanoparticles. Silver alloy is easier to melt than pure metal ingots because the impurities lower the melting point [17, 18, 19].”

We apologize for the mistake in converting the units.

Reviewer 3 Report

The review had a quite important improvement, and a consistent increase of references, and is good for a review...

nevertheless, some minor revisions are still needed.

1)I still have to disgaree with the phrase "Silver in the form of nanoparticles melts at 360°K while silver metal at 960°K."

as authors state in the response to my previous comment, melting point depends on dimensions, so the statement is without any sense if a dimension is not indicated!

 for example see Journal of Nanoscience and Nanotechnology, Volume 7, Number 11, November 2007, pp. 3805-3809(5)

so they have to clarify and define which are the AgNP melyting at 360 K, or at least the dimensional range of AgNP melting at that temperature!

2) the new paragraph added is a little bit messy, and also the english language has to be improved

phrases like "Despite the wide anisotropic and anti-inflammatory properties of AgNPsin the human body" should be corrected or clarified: what is an anoisotropic  property in the human body?

moreover, a phrase like

"That silver nanoprisms show a very high potential for antibacterial use in combination with color variation, which can give surfaces photothermal agents in teranostic applications -a unique feature -that can be used in cooperation with the classicalantibacterial action of silver nanoparticles"

is quite messy and seems to me whitout a precise sense in the present form

theranostic applications relates to diagnosys + therapy, as in the works of Astilean in ref 143, in which SERS imaging and phothermal tharpy are coupled. The other new references added (such 144 -150) relates on antibacteerial action given by AgNP based on silver ions release and, in some cases (when AgNP are nanoplates with LSPR absorption in the infrared region of the spectra) also able to give photothermal action, to obatin the killing of bacteria on bulk surfaces. They have nothing to do with theranostic!

so that part of test should be rewritten.

In general, a careful revison of phrases should be done in order to avoid misunderstandings.

Author Response

Comment: The review had a quite important improvement, and a consistent increase of references, and is good for a review...nevertheless, some minor revisions are still needed.

1)I still have to disgaree with the phrase "Silver in the form of nanoparticles melts at 360°K while silver metal at 960°K." as authors state in the response to my previous comment, melting point depends on dimensions, so the statement is without any sense if a dimension is not indicated! for example see so they have to clarify and define which are the AgNP melyting at 360 K, or at least the dimensional range of AgNP melting at that temperature!

Response: We are very grateful to the Reviewer for this valuable consideration. A detailed description of melting points with references is given in lines 115 to 119.

“Silver nanoparticles with a diameter of 2.4 nm melt at an approximate temperature 360°C, while unground silver without additives melts at 961.9 °C, however, boils at 2210°C [17, 18, 19]. In addition, its physical properties such as thermal, electrical and magnetic conductivity, also vary with the size of the silver nanoparticles. Silver alloy is easier to melt than pure metal ingots because the impurities lower the melting point [17, 18, 19].”

We apologize for the mistake in converting the units.

Comment: 2) the new paragraph added is a little bit messy, and also the english language has to be improved

Response: That fragment of text has been rewritten and English language corrected.

Comment: phrases like "Despite the wide anisotropic and anti-inflammatory properties of AgNPs in the human body" should be corrected or clarified: what is an anoisotropic property in the human body? moreover, a phrase like

"That silver nanoprisms show a very high potential for antibacterial use in combination with color variation, which can give surfaces photothermal agents in teranostic applications -a unique feature -that can be used in cooperation with the classicalantibacterial action of silver nanoparticles"  is quite messy and seems to me whitout a precise sense in the present form

Response: Both phrases in the sentences have been removed and the others rearranged in such a way that the description of the action of silver nanoparticles is more understandable.

Comment: theranostic applications relates to diagnosys + therapy, as in the works of Astilean in ref 143, in which SERS imaging and phothermal tharpy are coupled. The other new references added (such 144 -150) relates on antibacteerial action given by AgNP based on silver ions release and, in some cases (when AgNP are nanoplates with LSPR absorption in the infrared region of the spectra) also able to give photothermal action, to obatin the killing of bacteria on bulk surfaces. They have nothing to do with theranostic!

Response: The chapter on the description of the theranostic has been rebuilt in line with the reviewer's suggestions, for which I am very grateful.

 Once again thank you for your time to review our manuscript. I am very grateful for all comments and recommendations. I am open to further suggestions and look forward to hearing from you.

Sincerely,

Pawel Kowalczyk